# A 28 GHz Phased-Array Transceiver for 5G Applications in 22 nm FD-SOI CMOS

**DOI:** 10.3390/mi14051040

**Published:** 2023-05-12

**Authors:** Dan Cracan, Nourhan Elsayed, Mihai Sanduleanu

**Affiliations:** 1Nordic Semiconductor ASA, Swindon SN5 6NX, UK; dan.cracan@nordicsemi.no; 2GlobalFoundries Inc., Austin, TX 78735, USA; nourhan.elsayed1@globalfoundries.com; 3System on Chip Center, Khalifa University, Abu Dhabi P.O. Box 127788, United Arab Emirates

**Keywords:** phased array, transceiver, millimeter-wave, Doherty power amplifier

## Abstract

This paper presents the design and implementation of a 28 GHz phased array transceiver for 5G applications using 22 nm FD-SOI CMOS technology. The transceiver consists of a four-channel phased array receiver and transmitter, which employs phase shifting based on coarse and fine controls. The transceiver employs a zero-IF architecture, which is suitable for small footprints and low power requirements. The receiver achieves a 3.5 dB NF with a 1 dB compression point of −21 dBm and a gain of 13 dB.

## 1. Introduction

The development of fifth-generation (5G) wireless networks promises to deliver faster data transfer speeds, lower latency, and greater capacity compared to their predecessors. To fully realize the potential of 5G, however, new technologies and architectures are required. One such architecture is the phased array transceiver, which can support high-speed data transfer and beamforming for 5G applications [1,2,3].

Recently, zero-IF architectures have become of interest due to the trend toward higher integration [4,5,6]. Zero-IF architectures do not require IF filters and thus are prone to full integration on the chip. Moreover, there is only one LO signal, so the inherent reciprocal mixing is greatly reduced. The zero-IF architecture is a good candidate for a small footprint and lower power consumption receiver.

In this paper, we present the design and implementation of a 28 GHz phased array transceiver for 5G applications using 22 nm FD-SOI CMOS technology. The transceiver consists of a four-channel phased array receiver and a four-channel phased array transmitter, both operating at 28 GHz.

This paper is organized as follows: Section 2 presents the architecture of the phased array receiver, while Section 3 presents the architecture of the phased array transmitter; Section 4 presents the measurement results; and Section 5 draws conclusions.

## 2. Phased Array Receiver

A particularly interesting feature for 5G cellular receivers is the so-called beam-steering. This process involves combining multiple waves, using multiple antennas, so that they constructively interfere in a certain direction. To be able to control that direction, the phase/amplitude of each wave must be properly configured. This implies that a receiver will be able to scan for the optimum direction to increase the power of the received input signal, which is particularly important for waves at 28 GHz, which suffer from considerable attenuation while propagating through the atmosphere.

Figure 1 displays the block diagram of the proposed phase array receiver. The RF signal is fed to the LNA, which contains an input matching stage at the frequency band of interest around 28 GHz to attenuate potential out-of-band interfering signals. The pre-filtered and amplified RF signal is then converted to a differential form, amplified further with a variable gain amplifier (VGA), and then fed to a digitally controlled phase selector. Essentially, the phase selector provides one of the two phases of the RF signal at its output, based on a digital control signal. The phase selector would provide coarse control, the transmission line, which would be tunable, would provide fine control of the phase, and the active combiner would aggregate the outputs of multiple RF paths. Next, the RF signal is fed to I/Q mixers and down-converted to DC. As the mixers output a current signal, trans-impedance amplifiers are required for final processing with the low-pass filters.

The following subsections will provide more details on the key blocks of the receiver chain.

### 2.1. LNA and Active Balun

A one-stage LNA (Figure 2) was implemented. The input-matching circuitry at 28 GHz is also included. The L_1_, C_1_ tank in the drain of M_2_ resonates at 28 GHz to provide maximum gain at that frequency. The output of the LNA is fed, through a DC decoupling capacitor, to an active balun to convert the single-ended RF signal to a differential form.

### 2.2. Phase Rotator

Phase mismatch represents a potential source of error during signal down-converting. This can lead to incorrect demodulation of the data signal and corruption of the communication channel. Thus, to provide control of the LO phase, after the polyphase filter, a phase-rotating circuit is implemented (Figure 3). S_i_ and S_q_ are digital inputs responsible for coarse control of the phase. Fine control is achieved through I_1_ and I_2_ bias currents.

The LC tank is designed to resonate at 28 GHz to provide maximum gain at the LO frequency, and the values for the components are L = 158 pH and C = 61.2 fF. The bias currents I_1_ and I_2_ are set at 1 mA to allow for the large bandwidth necessary. Resistor R serves the purpose of improving the accuracy of the simple current mirror and is set at a rather low value of 147.5 Ω, to reduce the voltage drop across it, as the voltage headroom is limited, given that the input supply is at 1 V.

### 2.3. Tunable Transmission Line

Figure 4 shows the proposed structure of the tunable transmission line. Two metal structures have been added that can either be left floating or grounded. This will, respectively, impact a change in the inductance and capacitance of the transmission line and thus change the delay and, consequently, the phase of the signal passing through the transmission line. The inductance and capacitance controls must be changed simultaneously to maintain a constant characteristic impedance. The inductance and capacitance lines are driven by transistor switches. The transistor has been sized such that it offers a low resistance in the on state.

To allow for a binary weighted control of the phase, multiple sections, such as the one shown in Figure 4, are combined. A 7-section transmission line, configured in a 4-2-1 structure, has been simulated in Peakview, and a phase shift of 12° between the extreme control codes has been obtained.

### 2.4. Mixer

A zero-IF architecture was chosen for this application; thus, a 28 GHz mixer is required. An active topology was selected, as it provides conversion gain, as opposed to its passive counterpart. As the RF input is single-ended, a single-balanced active mixer was implemented, as shown in Figure 5.

A cascode load was implemented to provide for a higher impedance at the drains of transistors M_2_ to force most of the down-converted signal current to flow through the output capacitors C and thus increase the conversion gain. The RF stage is biased through a large resistance (R_3_ = 10 kΩ) to minimize signal leakage to ground, and the bias voltage is set at V_b,rf_ = 430 mV.

The LF stage is biased through 50 Ω resistors to match the output impedance of the phase rotator, and the bias voltage is set at V_b,lo_ = 600 mV. The output capacitors are set at 1 pF. Ideally, the capacitance should be as large as possible to provide a very low impedance for the down-converted signal, but that would translate to a very large area consumption.

### 2.5. Transimpedance Amplifier

As the mixer provides a current signal, a trans-impedance amplifier (TIA) is required to both convert the output signal to a voltage signal and amplify it, given that, typically, conversion gain is relatively low.

Figure 6 represents the circuit schematic of the TIA. Two feedback loops have been implemented: an internal loop consisting of M_2_, R_2_, inverter, and R_3_, and an external loop through R_4_ between the opposing input and output.

The purpose of the internal loop is to generate a high impedance at the drain of transistor M_1_ so that the gain of the stage is enhanced. The output feedback loop, which is negative, is employed for stability reasons. Resistors R_3_ have been chosen to be rather large to avoid leaking the RF signal to the ground. Resistors R_1_ are selected to provide an appropriate voltage at the input of the inverter so that the output DC common mode voltage is around half of the supply. The value of that voltage is determined by the bias current, selected at 550 μA, to provide enough bandwidth for the amplifier. The cascode current mirror requires a bias voltage that has been set at V_G2_ = 600 mV.

### 2.6. Low-Pass Filter

A low-pass filter is required after the TIA to eliminate unwanted frequency components, mainly due to LO injection. An all-pole topology Papoulis filter based on active unity gain buffers was adopted. The transfer function of the filter is given by Equation (1), where the assumption is made that the buffers display an infinite input impedance and zero output impedance:(1)$$H\left(s\right)=\frac{1}{1+sRC+s^{2}R^{2}C^{2}+\ldots s^{n}R^{n}C^{n}}$$

To provide a sharp roll-off, a seven-stage filter was implemented. To allow for flexibility, controls have been implemented to allow for filter bandwidth tuning. A standalone version of this filter, but with ten stages, was implemented, and results have been published in [7].

Using Equation (1), resistor and capacitor values can be identified sequentially; the first-order coefficient gives the value for R_1_C_1_, the second for R_1_C_1_R_2_C_2_. Resistors are set to the same value so that the unity gain buffers are presented with the same output/input load. The resistor and capacitor values for the filter are presented in Table 1.

To enable tuning of the filter frequency, two controls have been implemented. The coarse control switches on a parallel resistor, reducing the overall resistance and thus increasing the cut-off frequency. The fine control is implemented by a voltage-controlled MOS resistance. The range for the fine control is 0.7 V to 1 V. Figure 7 shows the block diagram of the filter.

Figure 8 shows the simulation results of the receiver.

## 3. Phased Array Transmitter

Figure 9 shows the block diagram of the 28 GHz phased-array transmitter. Input signals are coming from a fast digital-to-analog converter, and are then passed through a tunable low-pass filter to remove higher-order harmonics. To maintain the signal level at the mixer input within an acceptable range, a variable-gain amplifier is used. The mixer up-converts the baseband signal, which is then applied to four RF paths through an active power divider. Phase can be controlled finely via a tunable transmission line or coarsely via a phase selector.

Figure 10 depicts the architecture of the Doherty PA within one transmission pipe. It employs a main amplifier (Class-AB) and an auxiliary amplifier (switched cascode Class-E). A standalone version of this PA has been implemented and described in [8].

The phase-controlled input signal is equally split by an active balun that creates 180° out-of-phase signals. To provide phase balance between the two paths, a λ/4 transmission line with Z_0_ = 50 Ω is placed at the input and output of the main PA. To combine the two paths, a λ/4 transmission line with ≈35.4 Ω impedance is added. θ_c_ and θ_p_ serve the purpose of phase compensation.

In the following subsections, key blocks of the transmitter chain are presented.

### 3.1. Active Power Divider

Figure 11 shows the schematic of the power divider. Power division is implemented by splitting a current; however, further signal processing requires a voltage, and thus the divided current is converted back to a voltage. Transistor M_0_ converts the input voltage into a current, and transistors M_1_, M_2_, M_3_, and M_4_ convert the divided current back to a voltage. The power gain at each of the four outputs is:(2)$$G=10log\frac{P_{out}}{P_{in}}=20log\frac{V_{out}}{V_{in}}=20\log\frac{g_{m}R_{0}}{4},$$

### 3.2. Doherty PA with Delayed Switched Cascode Class E Amplifier

When the Class-E PA is off, the main PA is operating. This occurs when the input signal level is less than the threshold level of the auxiliary PA. The transmission line with Z_0_ impedance and a delay of θ_p_ provides an infinite impedance when seen from point X in Figure 12. For this, θ_p_ must satisfy the equation:
(3)$$Z_{0}\frac{\frac{1}{j\omega_{0}C_{0}}+jZ_{0}\tan\theta_{p}}{Z_{0}+\frac{\tan\theta_{p}}{\omega_{0}C_{0}}}=0,$$

The parallel capacitance of the auxiliary PA is C_0_. From Equation (3), θ_p_ results:(4)$$\theta_{p}=\mathrm{atan}\frac{1}{Z_{0}\omega_{0}C_{0}},$$

The impedance at the main PA is defined as:(5)$$R_{Main}=\frac{V_{1,Main}}{I_{1,Main}}=\frac{Z_{0}^{2}}{R_{Load}},P_{in}<P_{break},$$
where V_1,Main_ and I_1,Main_ are the fundamental components (voltage and current) of the main PA. The current at the breaking point is defined by I_break_ and I_1,Main_ is the maximum output current of the main PA I_1,Main._ The ratio of I_break_ to I_1,Main_ is given by:(6)$$\frac{I_{break}}{I_{1,Main}}=\frac{P_{break}-\cos\frac{\theta_{AB}}{2}}{1-\cos\frac{\theta_{AB}}{2}},$$

θ_AB_ is the conduction angle of the main PA. We enter the Doherty region as the input signal increases. Assuming that the main PA (Class-AB) has a constant voltage source for its maximum voltage (V_1,Main_), the load impedance at each PA is then defined as:(7)$$R_{Main}=\frac{V_{DD}-V_{k}}{I_{1,Main}},R_{Aux}=\frac{V_{DD}-V_{k}}{I_{1,Aux}},$$

The output voltage ($V_{Load}$) can then be calculated as:(8)$$V_{Load}=R_{Load}\left(I_{1,Main}+I_{1,Aux}\right)=R_{Load}I_{1,Main}\left(1+\frac{I_{1,Aux}}{I_{1,Main}}\right),$$
the following conditions should be satisfied:At point X, the output of both PAs should be in phase at the fundamental frequency. The phase compensation lines serve that purpose by adjusting θ_p_ and θ_c_.Maximum power transfer to the load from the auxiliary PA is achieved:
(9)$$\theta_{Y}+\theta_{Main}=\theta_{Z}+\theta_{Aux},$$

θ_Main_, and θ_Aux_ are the phases of the transmission function of the load network for each of the main and auxiliary PAs. The phases at points Y and Z in Figure 12 at the fundamental are referred to as θ_Y_ and θ_Y_. θ_Aux_ is calculated from Equation (9) after the output matching network. Accordingly, the load network of the main PA is optimized. The drain efficiency (η) of the Doherty PA can then be defined as:(10)$$\eta =\frac{P_{out}}{P_{DC}}=\frac{P_{out,Main}+P_{out,Aux}}{P_{DC,Main}+P_{DC,Aux}}=\frac{\eta_{E}P_{DC,E}+\eta_{AB}P_{DC,AB}}{P_{DC,E}+P_{DC,AB}},$$

## 4. Measurement Results

The transceiver was realized in the 22 nm CMOS FDSOI from GlobalFoundries. Figure 13 shows the measurement setup, and the chip photomicrograph is presented in Figure 14. The measurements were performed on a breakout DUT on the ELITE 300 probe station. For the measurement of the S-parameters and the noise figure, we used the Anritsu Vector Star ME-7838A VNA and the Anritsu MG3690C signal generator for the LO generation.

### 4.1. Receiver Chain

As can be seen in Figure 15, the receiver has a noise figure of 3.5 dB and a gain of around 13 dB at 28 GHz.

From Figure 16, it can be seen that the input 1 dB compression point of the receiver is about −21 dBm at 28 GHz.

Table 2 represents a comparison with the prior art. This design achieves the largest path gain with comparable noise figure values. This design also leads in terms of power consumption, mainly due to the zero-IF architecture and the fact that phase shifting occurs at RF, thus reusing the same down-conversion and filtering blocks for multiple RF paths. In terms of area, this design is larger than the one implemented in 40 nm CMOS, but this is probably because here we use large transmission lines to achieve fine-tuning of the phase, as opposed to the active counterparts, which have a smaller footprint.

### 4.2. Transmitter Chain

The PA was simulated and measured with different VGA settings (Figure 17). By changing the control inputs of the VGA, we can control the output gain of the PA. The peak measured gain is 17 dB at 28 GHz, and the saturated output power is 17.5 dBm. A frequency shift of 2 GHz is observed on the small signal parameters due to discrepancies in parasitic estimations.

A power sweep was performed on the PA using a spectrum analyzer. A saturated output power (Psat) of 17.5 dBm was measured (Figure 17). Two different measurement conditions were used to measure efficiency. One uses the auxiliary (Class-E) PA in the constant bias mode, and the second one uses the auxiliary PA in the switched mode (Figure 18 and Figure 19).

Under the constant bias condition, the Doherty PA peak PAE was 28% and showed a 3% degradation at back-off (25%), while the maximum drain efficiency measured was 48%. In the switched-mode condition. An improved peak (32%) and backed-off (31%) PAE was observed with a maximum DE of 59%. This is also an improvement in the overall PAE from the classical DPA architecture (Figure 20) [9].

Table 3 shows the comparison of the presented PAs to other state-of-the-art CMOS DPAs in the literature based on both Class-C and Class-E as the auxiliary PAs.

The results for using Class-E PA as the auxiliary show a substantial improvement in minimizing the degradation of PAE between the peak and back-off input power regions. It is also able to maintain a high output power while sustaining a high overall efficiency (PAE). The delayed switched-mode Class-E PA increases the peak PAE by 5% compared to the constant bias mode. In addition, the drop in efficiency from the peak to 6 db back-off region is decreased by 2% (from 3% to 1%). The figure of merit (FoM) from ITRS provides a performance metric that includes gain, output power, efficiency, and the operating frequency of an amplifier. This is used as a benchmark to compare the PAs in Table 2.

To the best of our knowledge, this design achieves the best FoM for peak and 6 dB back-off PAE compared to other Doherty PAs.

## 5. Conclusions

A 28 GHz four-phased array transceiver was implemented in 22 nm FD-SOI CMOS from Global Foundries and measured in this work. The receiver achieves a noise figure of 3.5 dB, a gain of 13 dB, and a 1 dB compression point of −21 dBm at the input. An active balun (on-chip) and a VGA precede the PA. Both the main and auxiliary amplifiers employ the stacked topology to obtain an increase in efficiency and output power. A gain of 17 dB and saturated output power of 17.5 dBm were reported. The auxiliary amplifier (Class-E) has two modes of operation: constant and switched-mode bias. The constant-bias Class-E DPA measured a peak PAE of 28% and a back-off PAE of 25%. The switched-mode Class-E DPA measured maximum PAEs of 32% and 31% at 6 dB back-off. A substantial improvement in PAE at both peak and 6-dB back-off is reported, along with the highest FoM compared to other state-of-the-art DPAs.

## Figures and Tables

**Figure 1 micromachines-14-01040-f001:**
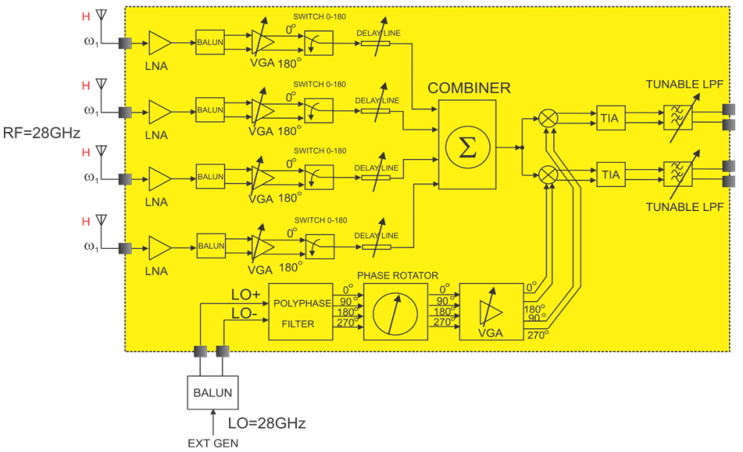
Block diagram of the receiver architecture.

**Figure 2 micromachines-14-01040-f002:**
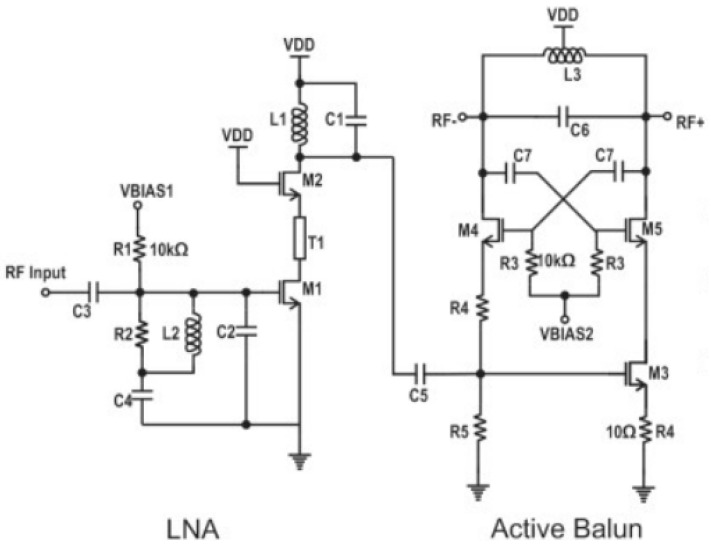
LNA followed by the active Balun.

**Figure 3 micromachines-14-01040-f003:**
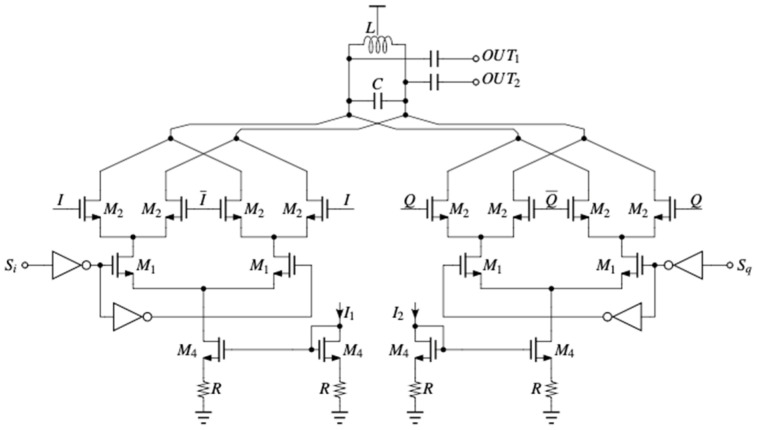
Circuit schematic of the phase rotator.

**Figure 4 micromachines-14-01040-f004:**
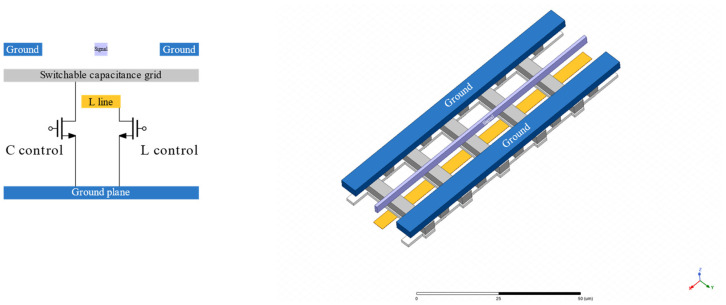
Top view and cross-section of one section of the tunable transmission line.

**Figure 5 micromachines-14-01040-f005:**
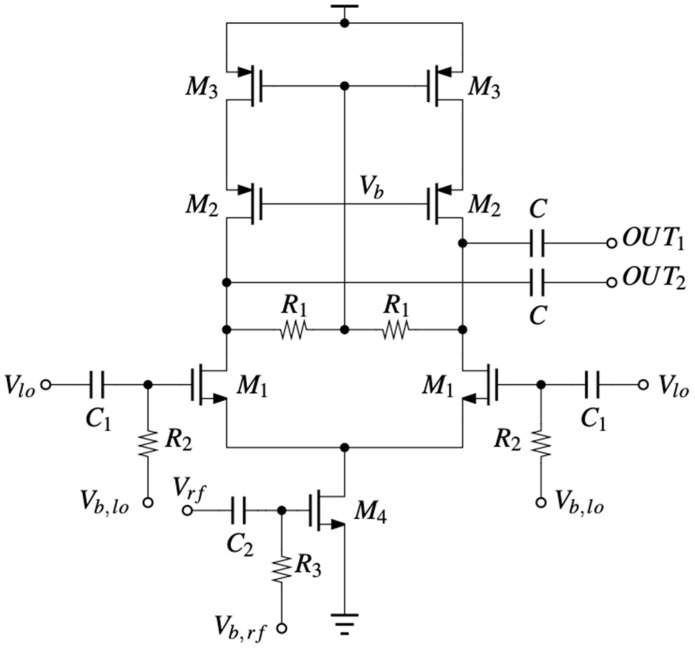
Circuit schematic of the mixer.

**Figure 6 micromachines-14-01040-f006:**
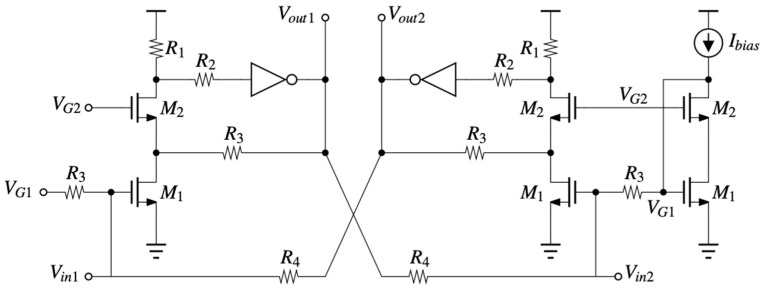
Circuit schematic of the trans-impedance amplifier.

**Figure 7 micromachines-14-01040-f007:**
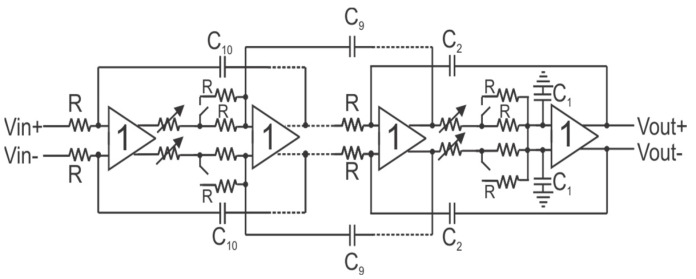
Tunable all-pole low-pass filter.

**Figure 8 micromachines-14-01040-f008:**
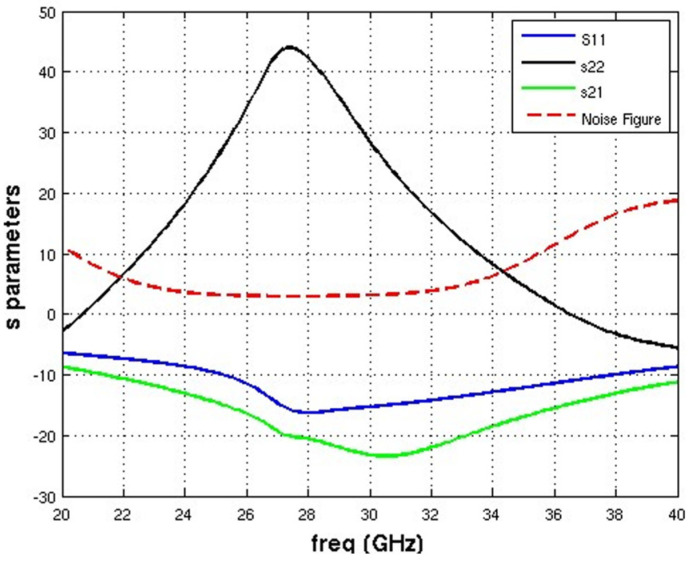
Simulation results for the receiver.

**Figure 9 micromachines-14-01040-f009:**
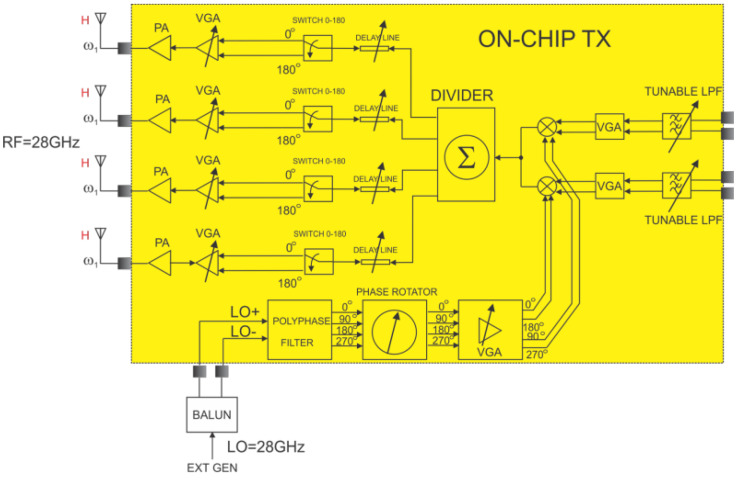
Block diagram of the phased array transmitter architecture.

**Figure 10 micromachines-14-01040-f010:**
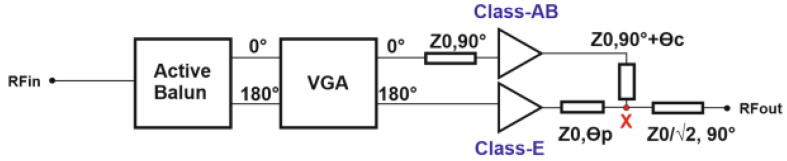
Block diagram of one transmitter pipe.

**Figure 11 micromachines-14-01040-f011:**
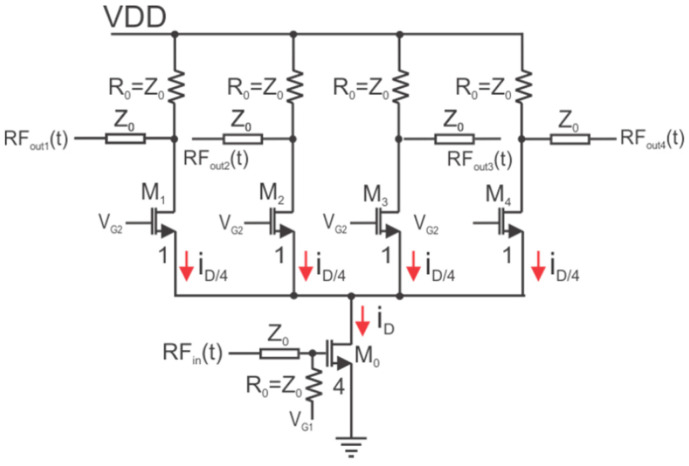
1:4 active power divider with cascoded outputs.

**Figure 12 micromachines-14-01040-f012:**
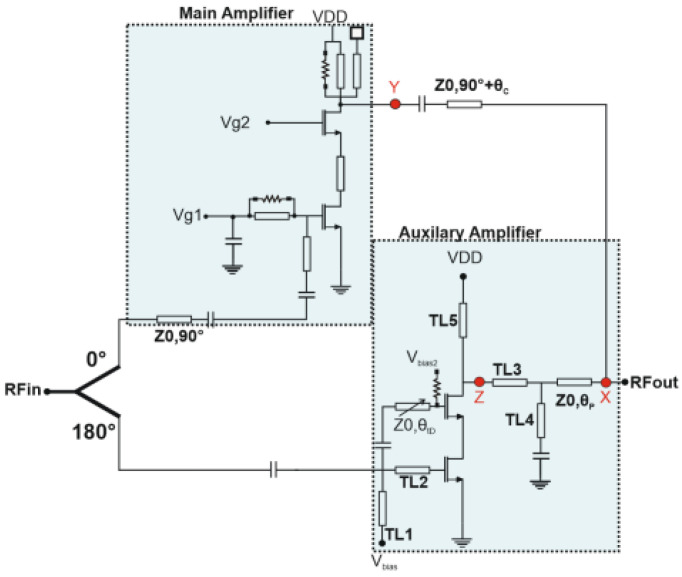
Doherty PA with Class-AB main and switched cascode class E auxiliary amplifiers.

**Figure 13 micromachines-14-01040-f013:**
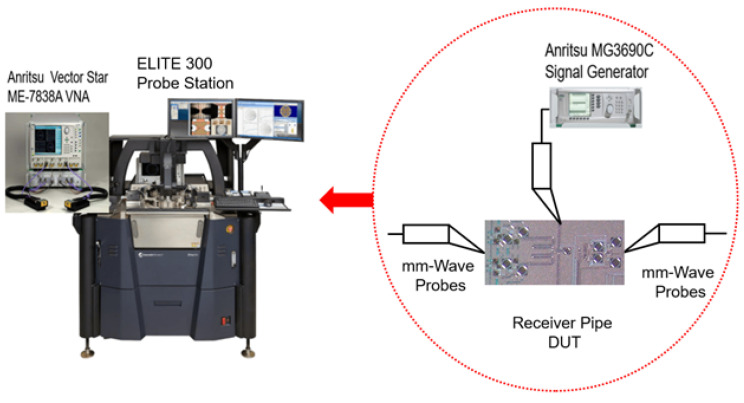
Measurement setup.

**Figure 14 micromachines-14-01040-f014:**
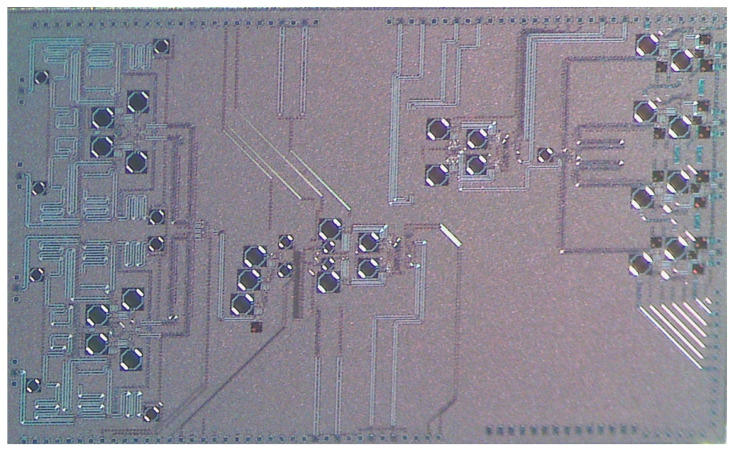
Four-phased array transceiver chip photomicrograph.

**Figure 15 micromachines-14-01040-f015:**
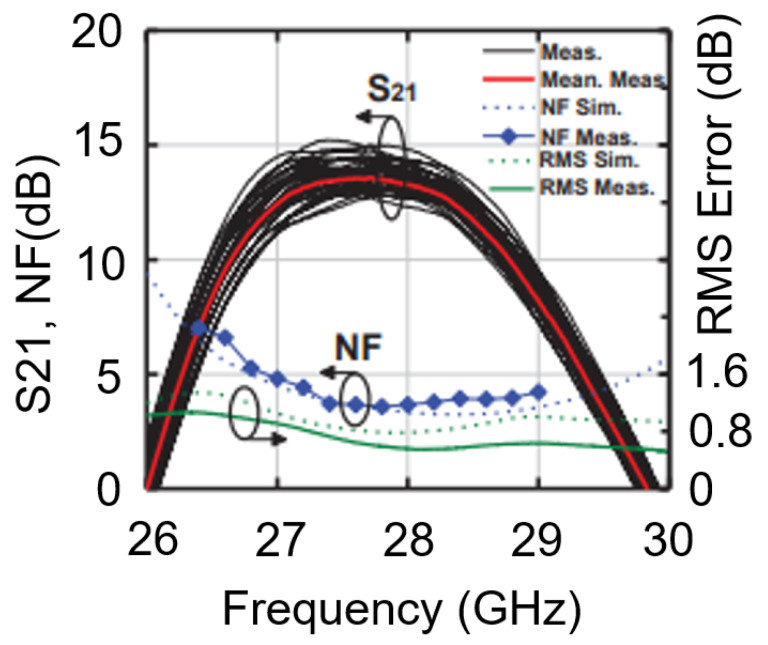
Receiver noise figure and gain (S_21_).

**Figure 16 micromachines-14-01040-f016:**
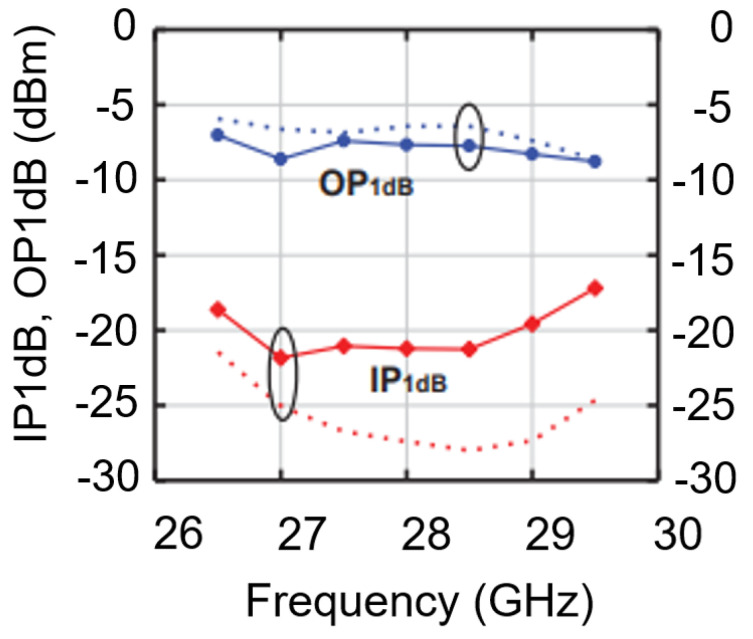
Receiver 1-dB input/output compression point.

**Figure 17 micromachines-14-01040-f017:**
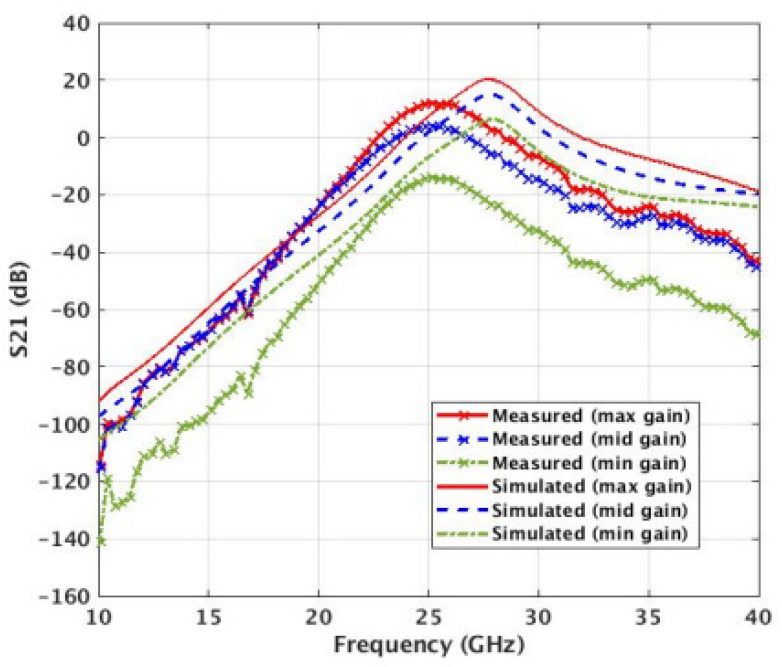
Simulated vs. measured S21 of the Balun + VGA + PA. Maximum measured gain of 17 dB. Controlling the gain of the VGA changes the maximum gain achieved by the DPA.

**Figure 18 micromachines-14-01040-f018:**
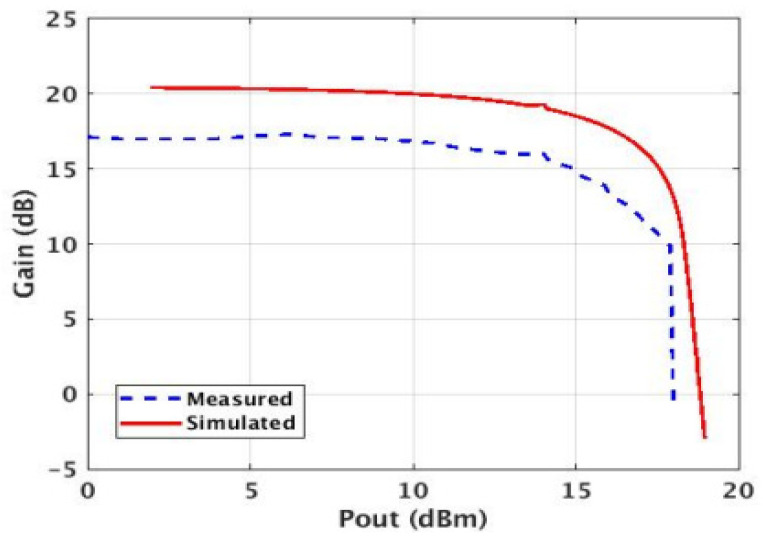
Simulated vs. measured power gain (power gain = 17 dB, Psat = 17.5 dBm).

**Figure 19 micromachines-14-01040-f019:**
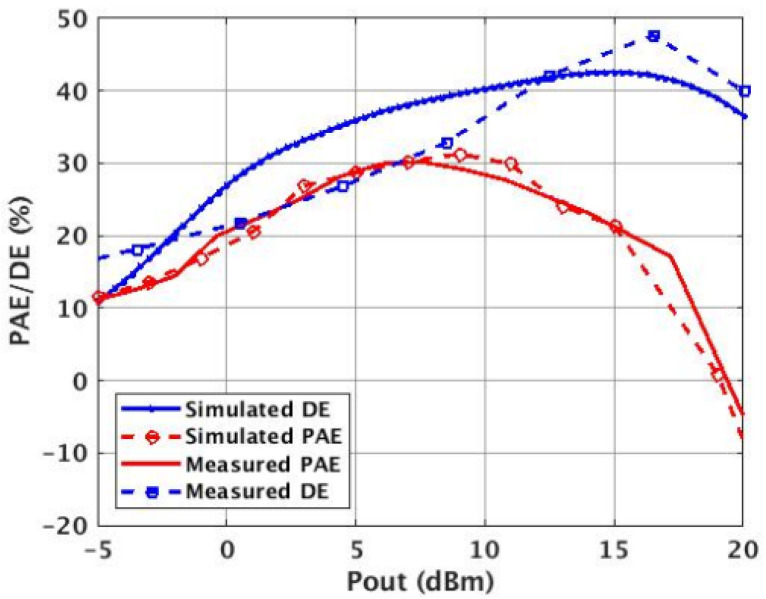
Measured and simulated efficiency (PAE/DE) of constant bias Class-E DPA. The measurements record a peak PAE of 28% and 25% at 6 dB back-off.

**Figure 20 micromachines-14-01040-f020:**
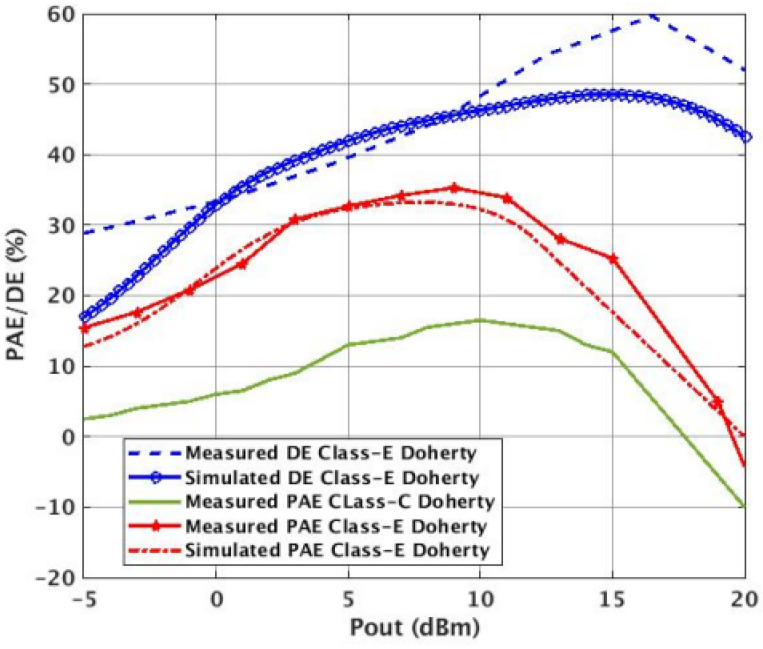
Measured and simulated efficiency (PAE/DE) of switched-mode Class-E DPA vs. measured PAE of a classical DPA. The measured DPA shows a 32% peak PAE and 31% at 6 dB back-off compared to the peak PAE of the classical DPA (16%).

**Table 1 micromachines-14-01040-t001:** Resistor (Ω) and capacitor (fF) values for the 7-stage all-pole filter.

R	C1	C2	C3	C4	C5	C6	C7
700	7.8	15.7	23.9	32.6	43	53.8	70.9

**Table 2 micromachines-14-01040-t002:** Benchmark and comparison with other phased array receivers.

	This Work	[4]	[5]
Technology	22 nm CMOS	130 nm SiGe BiCMOS	40 nm CMOS
Phase shifter	Hybrid	Passive	Active
Array size	4	32	4
Frequency (GHz)	28	28	15
Single path gain (dB)	14–30	34	23
NF (dB)	3.5	3.7	3.4
1 dB compression point (dBm)	−21	−22.5	−37
Power consumption (mW)	330	3300	463
Chip area (mm^2^)	8.1	165.3	1.8

**Table 3 micromachines-14-01040-t003:** Benchmark and comparison with other power amplifiers.

Ref	Tech.	Freq. (GHz)	Psat (dBm)	Peak PAE (%)	BO PAE (%)	Gain (dB)	FoM *	FoM (BO)	Matching Network	Architecture
This work	22 nm FDSOI	28	17.5	28	25	17	29.3	28.7	On-chip	Doherty Class-E
This work	22 nm FDSOI	28	17.5	32	31	17	29.7	29.5	On-chip	Doherty Class-E Switched mode
[10]	40 nm CMOS	2.5	17.5	34	25	29	25.9	19.6	Off-chip	Doherty Class-E Digital control
[11]	45 nm SOI	42	18	23	17	7	19.4	18.1	On-chip	Doherty
[12]	40 nm CMOS	77	16.2	12	5.7	9	25.7	22.5	On-chip	Doherty Transformer based
[13]	0.13 μm CMOS	60	7.8	3	1.5	13.5	19.3	16.3	On-chip	Doherty
[14]	45 nm SOI	14	22	24	20	8	15.1	14.3	On-chip	Series Doherty
[15]	90 nm CMOS	71–76	11.7	30.6	15.6	4.7	22.8	19.9	On-chip	Doherty
[16]	45 nm SOI	28	22.4	40	28	10	26.5	24.9	On-chip	Doherty

* FoM = P_sat_(dBm) + Gain(dB) + 10 log_10_(Peak/BO PAE) + 20 log_10_f_o_/f_max_.

## Data Availability

The authors confirm that the data used in this study is either experimentally extracted and provided throughout the article or referenced below.

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
