# Peer review of "A 28 GHz Phased-Array Transceiver for 5G Applications in 22 nm FD-SOI CMOS"

_micromachines, 2023, doi:10.3390/mi14051040_

Round 1
Reviewer 1 Report
This work presented a 5G transceiver design at 28GHz in 22nm FD-SOI CMOS process. It employs a Zero-IF architecture for small footprint and low power requirements. It shows a 3.5dB NF and a -21dBm 1dB compression point and a 13dB gain. The paper can be accepted for publication after addressing the following questions and comments:
1. The authors should double check English grammar across the paper. For instance, in Line 33, "while Section 3 of the phased array" should be "while Section 3 presents the architecture of the phased array". In Line 179, "to" was missed after “in order”. In Line 234, "is" was missed after "This". Besides, in Conclusions, "with" was missed after "on-chip" in Line 4.
2. The authors should add the simulation results of their tranceiver design in Section 2 and 3.
3. It is not clear how the prototype was characterized in lab. The authors should present more details on the measurement in Section 4. For instance, how were the gain and PAE measured in the lab?
Author Response
The paper has been proof read with grammarly. Some simulation results have been added. Measurement setup has been added
Reviewer 2 Report
This manuscript presents the design and implementation of a 28GHz phased array transceiver for 5G applications, utilizing 22-nm FD-SOI CMOS technology. Please find the comments below.
1) The contribution of this paper is not clear in the introduction section. The reviewer recommends adding more information to clarify it. Additionally, more reference reviews are needed in the introduction section.
2) The references are disorganized and incorrect. For example, the references listed in the table 3 are wrong. The reviewer recommends thoroughly reviewing all references in the manuscript.
3) There are a few typos:
Line 179: “phase signals in order avoid having an off-chip splitter.”
Line 250: “As can be seen in Figure 13 shows, the receiver has noise figure of 3.5 dB and gain of”
Line 262: “Table 1 represents a comparison with prior art.” It should be Table 2, not Table 1.
Author Response
More references have been added to the introduction. References have been corrected. Typos have been corrected.